# Gene Expression Profiling of Olfactory Neuroblastoma Helps Identify Prognostic Pathways and Define Potentially Therapeutic Targets

**DOI:** 10.3390/cancers13112527

**Published:** 2021-05-21

**Authors:** Chiara Romani, Eliana Bignotti, Davide Mattavelli, Anna Bozzola, Luigi Lorini, Michele Tomasoni, Laura Ardighieri, Vittorio Rampinelli, Alberto Paderno, Simonetta Battocchio, Cristina Gurizzan, Paolo Castelnuovo, Mario Turri-Zanoni, Carla Facco, Fausto Sessa, Alberto Schreiber, Marco Ferrari, Antonella Ravaggi, Alberto Deganello, Piero Nicolai, Michela Buglione, Davide Tomasini, Roberto Maroldi, Cesare Piazza, Stefano Calza, Paolo Bossi

**Affiliations:** 1Angelo Nocivelli Institute of Molecular Medicine, ASST Spedali Civili di Brescia, University of Brescia, 25123 Brescia, Italy; chiara.romani@unibs.it (C.R.); eliana.bignotti@asst-spedalicivili.it (E.B.); antonella.ravaggi@unibs.it (A.R.); 2Department of Medical and Surgical Specialties, Radiological Sciences and Public Health, University of Brescia, 25123 Brescia, Italy; davide.mattavelli@unibs.it (D.M.); m.tomasoni022@unibs.it (M.T.); alberto.deganello@unibs.it (A.D.); ceceplaza@libero.it (C.P.); 3Unit of Otorhinolaryngology–Head and Neck Surgery, ASST Spedali Civili di Brescia, 25123 Brescia, Italy; v.rampinelli@unibs.it (V.R.); a.paderno006@unibs.it (A.P.); alberto.schreiber@unibs.it (A.S.); 4Department of Pathology, ASST Spedali Civili of Brescia, 25123 Brescia, Italy; anna.bozzola@gmail.com (A.B.); Lauraardighieri@gmail.com (L.A.); simonetta.battocchio@asst-spedalicivili.it (S.B.); 5Medical Oncology Unit, Department of Medical and Surgical Specialties, Radiological Sciences and Public Health, ASST Spedali Civili of Brescia, University of Brescia, 25123 Brescia, Italy; luigilorini91@gmail.com (L.L.); c.gurizzan@unibs.it (C.G.); 6Division of Otorhinolaryngology, Department of Biotechnology and Life Sciences, University of Insubria, 21100 Varese, Italy; paolo.castelnuovo@uninsubria.it (P.C.); mario.turrizanoni@asst-settelaghi.it (M.T.-Z.); 7Head and Neck Surgery & Forensic Dissection Research Center (HNS&FDRc), Department of Biotechnology and Life Sciences, University of Insubria, 21100 Varese, Italy; 8Department of Pathology, ASST Sette Laghi, University of Insubria, 21100 Varese, Italy; carla.facco@ospedale.varese.it (C.F.); fausto.sessa@uninsubria.it (F.S.); 9Section of Otorhinolaryngology–Head and Neck Surgery, Department of Neurosciences, University of Padua, 35121 Padua, Italy; marco.ferrari@unipd.it (M.F.); piero.nicolai@unipd.it (P.N.); 10Department of Radiation Oncology, ASST Spedali Civili di Brescia, University of Brescia, 25123 Brescia, Italy; michela.buglione@unibs.it (M.B.); davide.tomasini@unibs.it (D.T.); roberto.maroldi@unibs.it (R.M.); 11Unit of Biostatistics and Bioinformatics, Department of Molecular and Translational Medicine, University of Brescia, 25123 Brescia, Italy; stefano.calza@unibs.it; 12BDbiomed, Big and Open Data Innovation Laboratory, University of Brescia, 25123 Brescia, Italy

**Keywords:** olfactory neuroblastoma, gene expression, molecular profiling, prognosis

## Abstract

**Simple Summary:**

The gene expression profile of ONB defines a group of patients with a dismal prognosis and identifies potentially targetable pathways. Better prognostic stratification may offer new tailored approaches for the treatment and follow-up of ONB. The integration of new therapeutic agents with standard surgical and RT strategies may improve the outcomes in cases with worse prognoses. Furthermore, the ontogenesis of ONB in basal and neural subtypes is mirrored by different transcriptional pathways, paving the way towards different therapeutic approaches.

**Abstract:**

Olfactory neuroblastoma (ONB) is a rare sinonasal neoplasm with a peculiar behavior, for which limited prognostic factors are available. Herein, we investigate the transcriptional pathways altered in ONB and correlate them with pathological features and clinical outcomes. We analyze 32 ONB patients treated with curative intent at two independent institutions from 2001 to 2019 for whom there is available pathologic and clinical data. We perform gene expression profiling on primary ONB samples and carry out functional enrichment analysis to investigate the key pathways associated with disease-free survival (DFS). The median age is 53.5 years; all patients undergo surgery and a pure endoscopic approach is adopted in the majority of cases (81.2%). Most patients have advanced disease (stages III–IV, 81.2%) and 84.4% undergo adjuvant (chemo)radiotherapy. The median follow-up is 35 months; 11 (26.8%) patients relapse. Clinical characteristics (gender, stage and Hyams’ grade) are not associated with the outcomes. In contrast, TGF-beta binding, EMT, IFN-alpha response, angiogenesis, IL2-STAT5 and IL6-JAK-STAT3 signaling pathways are enriched in patients experiencing recurrence, and significantly associated with shorter DFS. Clustering of transcriptional profiles according to pathological features indicates two distinct molecular groups, defined by either cytokeratin-positive or -negative immunostaining. Definition of the characterizing ONB transcriptomic pathways may pave the way towards tailored treatment approaches.

## 1. Introduction

Olfactory neuroblastoma (ONB), also named esthesioneuroblastoma, is a rare sinonasal neoplasm of the olfactory neuroepithelium [1] that is characterized by peculiar biological behavior. Its close relationship with the cribriform plate accounts for its tendency to intracranial extension and brain invasion. Cervical metastases are not frequently found at presentation, but, in the long-term, may occur in 20–25% of patients [2,3]. Similarly, the incidence of distant metastases at presentation is low but can reach 25% during long-term follow-up [4]. 

Historically, parameters related to tumor extension, such as the Kadish staging system and Dulguerov classification, have been considered as the main prognostic factors, together with nodal and metastatic status. Nevertheless, both Kadish and Dulguerov classifications are poorly informative of tumor biologic aggressiveness, and their prognostic power has been put into question by several papers [5]. Recently, Hyams’ histological grade has emerged as a major prognosticator [6]. This is a four-tiered staging system based on histopathological features, although it is usually simplified in low (I–II) vs. high (III–IV) grades. This subdivision depicts the dual nature of ONB, as it can segregate two groups of tumors with different biological behaviors. Low-grade lesions tend to be indolent, with limited local aggressiveness and scarce propensity to nodal or distant spread, while showing excellent outcomes and prolonged survival. In contrast, high-grade tumors usually present with large and invasive masses and have a higher propensity for distant metastasis and early recurrence after treatment, which all translate into a poorer short- and long-term prognosis [7]. In recent years, Hyams’ grade has been considered a critical factor even in treatment planning, since high-grade tumors (in particular, grade four), due to their metastasizing propensity, may benefit from more intensive treatments, while low-grade lesions are usually managed by less aggressive therapeutic protocols. 

However, even Hyams’ grading system exhibits some shortcomings. The most relevant is that it is a subjective classification. Moreover, its attribution is particularly challenging on the pure basis of a limited preoperative biopsy, which could be poorly representative of the entire tumor. As a consequence, major clinical decisions on treatment planning (i.e., induction chemotherapy vs. upfront surgery) may rely on biased data.

Recent evidence suggests that multi-omics analysis can provide a deeper understanding of such a complex pathology. In fact, even though molecular studies of ONBs are scarce due to the intrinsic rarity of this histotype, only a few recurrent genomic aberrations or somatic mutations in known cancer genes [8,9], with limited predictive and prognostic impact, have been consistently reported. More recently, transcriptional analysis of ONBs through RNA sequencing contributed to describing two subgroups of tumors related to cell ontogeny and associated with distinct clinicopathological features [10], reinforcing the translational relevance of gene expression profiling approaches.

Herein, we present an analysis of the transcriptional pathways altered in ONB and their correlation with pathological features and clinical outcomes.

## 2. Materials and Methods

### 2.1. Patient Information

All patients included in the study were diagnosed and treated with curative intent for ONB in the Otorhinolaryngology and Head and Neck Surgery units of ASST Spedali Civili, University of Brescia, and ASST Sette Laghi, University of Insubria, Varese, Italy, from 2001 to 2019. Exclusion criteria were: (a) presence of distant metastasis at diagnosis; (b) surgery performed as salvage treatment after local recurrence; (c) absence of clinical/pathological and/or follow-up data; (d) formalin-fixed paraffin-embedded (FFPE) tumor blocks not available for RNA extraction.

Written informed consent was obtained from all patients enrolled as per the protocol approved by the Research Review Board of the ASST Spedali Civili, University of Brescia, Brescia, Italy (study reference number: NP-3802).

### 2.2. Tissue Collection and Histopathological Analysis 

FFPE blocks from ONB tissue were generated at the time of primary surgery from radiotherapy (RT)-naïve patients. All cases were reviewed by two expert head and neck pathologists (S.B. and L.A.) and the diagnosis confirmed using hematoxylin and eosin staining. In case of disagreement about grading between the two pathologists, a final grade was reached by consensus.

All immunohistochemical analyses were performed on serial 5 µm sections of a selected FFPE representative tumor block. The markers used for diagnostic purposes were chromogranin A (clone LH2H10, DB-Biosystem, Franklin Lakes, NJ, USA), synaptophysin (clone DAK-SYNAP, Dako, Glostrup, Denmark), S100 protein (polyclonal, Leica, Wetzlar, Germany), pan-cytokeratin (clone MNF116, Dako, Glostrup, Denmark) and Ki-67 proliferation index (clone 30-9, Ventana, AZ, USA). Immunohistochemical analysis was performed on the entire slide of the selected block and the percentage of positive neoplastic cells was determined. Quantification of the Ki-67 index was done by counting the positive neoplastic cells among all tumor cells present in the area of interest (hotspot area) and by determining the percentage of positive neoplastic cells in the entire tumor area.

### 2.3. RNA Extraction

Total RNA was isolated with an AllPrep DNA/RNA FFPE Kit (Qiagen, Hilden, Germany) according to the manufacturer’s instructions. RNA concentration and 260/280 absorbance ratio (A260/280) were measured with the Infinite M200 spectrophotometer (TECAN, Mannedorf, Switzerland). Quality control of extracted RNA was assessed with RT-qPCR for 18S rRNA (ThermoFisher Scientific, Waltham, MA, USA).

### 2.4. GENE Expression Profiling

To generate gene expression profiles, total RNA was converted to cRNA and then to cDNA that was subsequently hybridized, according to the manufacturer’s guidelines, on a Human Clariom S GeneChip (ThermoFisher Scientific), optimized for FFPE samples and able to accurately measure the gene-level expression of >20,000 well-annotated genes. A 3000 7G Scanner (ThermoFisher Scientific) was used in conjunction with GeneChip Operation Software (ThermoFisher Scientific) to generate a single CEL file for each hybridized cDNA.

### 2.5. Microarray Data Processing and Analysis

Arrays underwent quality control before being processed with the RMA algorithm. A potential batch effect (center of data processing) was removed using a linear model [11]. Hierarchical clustering was performed using a Euclidean distance metric and Ward’s agglomerative algorithm [12]. 

Gene filtering based on association with clinical features was performed using individual-level models. Disease-free survival (DFS) was evaluated using Cox proportional hazards models, and genes were filtered using unadjusted *p*-values of less than 5%. Differential expression (DE) was modeled using linear models with t-statistic computation based on an Empirical Bayesian algorithm [11]. Filtered features were further processed to evaluate potential pathways or gene set enrichment using Gene Set Enrichment Analysis (GSEA) [13] and both Hallmark [14] and GO gene-set lists.

An enrichment score for selected gene sets (hereafter, a global score) for each patient was computed using single-sample GSEA (ssGSEA) [15]. ssGSEA computes an enrichment score for every subject gene-set based on a non-parametric approach: the score can be interpreted as the level of global expression of a specific gene-set for each subject.

### 2.6. Statistical Analysis

Data were described using the mean (standard deviation, SD) and median (interquartile range, IQR) for quantitative variables and percentages for categorical variables.

Overall survival (OS) and DFS were modeled as a function of clinical variables using Cox proportional hazard models with the log-rank test. The follow-up distribution was estimated using the Kaplan-Meier method. Optimal cut-off points for continuous variables, as predictors of time-to-event outcomes (OS, DFS), were computed using maximally selected log-rank statistics [16]. All tests were two-sided and assumed a significance level of 5%. All analyses were performed using R (version 4.0.3, https://www.r-project.org/, accessed on 1 April 2021).

## 3. Results

### 3.1. ONB Cohort Description

A total of 32 patients were included in the study. Demographic, clinical, pathological and treatment-related characteristics are outlined in Table 1. The median age was 53.5 years (range 18–78) with a slight male prevalence (53.1%). All patients underwent surgery; in one case (3.1%), induction chemotherapy was administered. Transnasal endoscopic (71.8%) or combined cranio-endoscopic (9.4%) craniectomies were the most common procedures performed; in three (9.4%) cases, a more conservative endoscopic resection without craniectomy was feasible. The surgical margins were negative at the final histology in 70.4% of the cases. 

Most patients presented with advanced disease (stages III–IV, 81.2%). Disease staging strictly reflected local extension (pT), since in one case only (3.1%), nodal metastasis was observed (pN2b). Almost all patients (84.4%) underwent adjuvant treatments while just one patient received induction chemotherapy. Adjuvant RT and combined chemotherapy and RT (ChRT) were administered in 68.8% and 15.6% of cases, respectively. The recurrence rate was 20% in patients not receiving adjuvant treatment vs. 15% in patients receiving adjuvant RT.

### 3.2. Pathological Data

The morphological re-evaluation showed that most patients were affected by Hyams’ grade II (50.0%) and III (37.5%) tumors; less commonly, grades I and IV were recorded (3.1% and 9.4%, respectively). No change in Hyams’ grade was applied, except for three cases that were upgraded from grade III to IV. Overall, an equal distribution between low (I–II, 53.1%) and high (III–IV, 46.9%) Hyams’ grades was observed. As detailed in Table 2, all cases showed the expression of neuroendocrine markers chromogranin A and synaptophysin. Focal positivity, defined by less than 10% of neoplastic cells, was observed in a single high-grade tumor. Pan-cytokeratin expression was observed in 8/32 (25%) cases, seven of which were high-grade tumors (III–IV Hyams’ grade). The S100 protein revealed a preserved and almost complete rim of sustentacular cells at the periphery of tumor nests in 21/32 (65.6%) cases, while 10/32 (31.2%) showed markedly decreased S100-positive staining with sparse positive cells, still mostly distributed along the periphery or inside the tumor nests. Only one case of Hyams’ IV grade tumor (3.1%) was completely negative. Overall, the average (computed as geometric mean) Ki-67 proliferation index was 12.5% (SD, 3.23%), with a slight increase in high-grade tumors (III–IV 12.5% vs. I–II 11.6%).

### 3.3. Oncologic Outcomes

The median follow-up was 35 months (IQR, 39; range, 5–172). Overall, three deaths (9.4%) were recorded, among which two (6.2%) were cancer-related, while 11 (34.4%) patients experienced disease recurrence (two local, three nodal and one distant recurrence). The median disease-free interval for recurrent patients was 51 months (IQR, 45; range, 8–99). 

No significant difference in terms of OS or DFS was observed according to Hyams’ grade (III–IV vs. I–II, DFS *p* = 0.343, OS *p* = 0.377), tumor stage (III–IV vs. I–II, DFS *p* = 0.355, OS *p* = 0.476), age (≥ 54-year-old vs. <54 years, DFS *p* = 0.838, OS *p* = 0.700) or gender (male vs. female, DFS *p* = 0.900, OS *p* = 0.884).

### 3.4. Pathways Associated with Poor DFS

To identify gene signatures that could predict relapse, 32 primary ONB cases were profiled for mRNA expression using Affymetrix Clariom S microarrays. The top genes selected based on DFS association (FC > 1.5 and FC < 0.67, *p* < 0.05, Figure 1A, Appendix A) were included in GSEA: patients characterized by poor DFS had disease showing enrichment in pathways related to TGF-beta binding, epithelial-mesenchymal transition (EMT), UV response, allograft rejection, IFN-alpha response, angiogenesis, IL2-STAT5 and IL6-JAK-STAT3 signaling (FDR < 0.25, *p* < 0.05) (Figure 1B, Appendix A). 

The association between deregulated pathways and DFS in ONB patients was evaluated using a Kaplan-Meier estimator and log-rank test. The pathway-specific global score, computed using the ssGSEA algorithm, was dichotomized using maximally selected statistics and the 32 ONB samples were classified accordingly into two groups, labeled as high/low score. As shown in Figure 1C, the DFS of the high-score group was worse than that of the low-score group, indicating the potential prognostic relevance of these signatures.

### 3.5. Molecular Variants

According to the molecular-based ONB subgroups classification proposed by Classe et al. [10], two major subtypes with either a neural or basal transcriptional signature can be identified based on the expression of selective markers. Specifically, basal ONBs are high-grade tumors showing positive immunostaining for cytokeratins and increased Ki-67 compared to neural ONB, which are well-differentiated tumors with high expression of chromogranin A and synaptophysin, and S100-positive sustentacular cells. In our cohort of primary ONB, expression of cytokeratins was observed in eight (25.0%) cases (hereafter referred to as CK^positive^), mainly represented by advanced-stage tumors (46.7% stages III–IV vs. 5.9% stages I–II, *p*-value 0.013, Fisher’s exact test). Compared to others, CK^positive^ samples showed a significant increase in the Ki-67 proliferation index (28.1% vs. 9.3%, *p*-value = 0.009) and a lower density of S100-positive sustentacular cells (25.0% vs. 70.8%, Fisher’s exact test *p* = 0.038). Together these data are consistent with the basal vs. neural ONB subtypes distinction. 

We then compared the gene expression profiles of ONB patients, grouped by IHC staining, into CK^positive^ or CK^negative^ (Figure 2A) to gain insight into the molecular pathways that drive putative basal or neural differentiation. Overall, 1777 genes were identified as differentially expressed between the two IHC subgroups, of which 1015 were upregulated and 762 were downregulated in CK^positive^ samples (unadjusted *p*-value < 0.05, Figure 2B, Appendix A). The selected genes were used for a GSEA analysis using either GO or Hallmark gene-set lists. Thirty GO gene sets were significantly enriched (FDR < 25%) in the CK^positive^ samples, with the top four being related to olfactory receptor activity, sensory perception of smell and chemical stimulus (all *p* < 0.001). Furthermore, E2F targets, MYC targets and KRAS hallmark pathways were associated with this phenotype, emphasizing a different nature of this class compared to CK^negative^ samples, characterized by a positive enrichment of gene sets related to oxidative phosphorylation, ATP synthesis, and complexes of the respiratory chain in the mitochondrial inner membrane, and pathways related to microtubule function and regulation, including TAU microtubules binding protein (Figure 2C, Appendix A). At the single gene level, budding uninhibited by benzimidazoles 1 (BUB1) was the most significantly upregulated gene in CK^positive^ tumors (FC < 1.5, *p* < 0.05)

## 4. Discussion

ONB is a rare sinonasal cancer, with heterogeneous clinical behavior, ranging from indolent forms to rapidly growing disease, mainly metastasizing to the lymph nodes, brain and lungs. The rarity of the disease and the presence of small retrospective series explain the paucity of well-known prognostic factors and the inherent difficulty in defining personalized treatment approaches. Late recurrences are not uncommon, and can often be successfully managed by salvage attempts such as surgery and RT. In this regard, DFS represents a more reliable endpoint when aiming to define prognostic factors in such a rare disease.

In this study, we performed global gene expression profiling of a large cohort of primary ONB samples to fully characterize the ONB transcriptome and identify relevant pathway alterations associated with DFS that may be therapeutically exploited. We found key biological processes, including angiogenesis and EMT, which were significantly enriched in patients experiencing recurrence, thus offering potentially druggable pathways. Targeting angiogenesis by blocking one or multiple axes (i.e., VEGF, HGF, FGF) could be considered, in the near future, as an alternative strategy for ONB treatment, as already underlined in previous small experiences [8]. EMT has been associated with a worse prognosis in head and neck squamous cell carcinoma, being linked to tumoral immune escape pathways [17]. Highly correlated with EMT, the TGF-beta pathway was significantly upregulated in patients with poor DFS in our analysis, thus confirming its well-known role as the primary inducer of EMT in different cancer types [18]. This pathway could also offer potential future opportunities for personalized therapy since several agents targeting its components are currently being developed and evaluated in clinical trials [19]. Moreover, TGF-beta signaling acts as an important suppressor of the adaptive and innate immune responses during tumor progression, and is a mechanism of direct immunosuppression, as well as through the EMT and angiogenesis pathways [20]. In this regard, the immune-excluded profile of these tumors would not benefit from immune checkpoint inhibitors alone, but the concomitant inhibition of TGF-beta could potentially restore the ability of the immune system to block tumor growth.

As already reported in many types of cancer, IL-6/JAK/STAT3 signaling is also significantly hyperactivated in the subset of ONB patients with a poor clinical prognosis. This pathway has either a key role in promoting tumor invasive growth and metastasis or a strong involvement in silencing the antitumor immune response. Therefore, targeting the IL-6/JAK/STAT3 pathway in patients with ONB might provide therapeutic benefit by directly suppressing tumor cell proliferation and restoring antitumor immunity, as already demonstrated in several other histotypes [21]. Combination therapy with IL-6/JAK/STAT3 signaling axis inhibitors and immune checkpoint inhibitors could, therefore, be considered due to their potential synergistic therapeutic effects. 

Lately, the IFNα pathway was also seen to be significantly enriched in patients with shorter DFS. Canonically, IFNα has been considered a tumor-suppressor molecule with the pro-inflammatory property of recruiting dendritic cells and activating T cells. However, if IFNα signaling becomes chronically activated, it induces increased tumor PDL-1 expression along with dendritic cell exhaustion, thus contributing to immune evasion and supporting a pro-tumorigenic microenvironment [22].

In our analysis, we also evaluated the transcriptomic characteristics in relation to the ONB variants described in the literature. Results of pathological evaluation of ONB samples according to the expression levels of CK and Ki-67 support the existence of two distinct subgroups, basal and neural, concordant with that described by Classe and colleagues [10]. Compared to the neural subtype, basal ONB is confirmed to be less differentiated, more advanced and with an increased proliferation index. This points to a potential prognostic role of CK and Ki-67 that remains to be established in large cohorts with more informative follow-up. Additionally, this ontogenesis-related molecular classification links the transcriptional profile of both subtypes to the two principal cell types of the olfactory epithelium, the globose basal cell progenitor and the more differentiated olfactory sensory neuron, respectively, in agreement with the cancer stem cell (CSC) paradigm [23,24]. Consistent with this, functional analysis indicates critical transcriptional programs enriched in basal ONB compared to the neural, including Myc and E2F, two closely interrelated pathways that play a central role in cell proliferation control and cell fate decision [25,26]. Notably, in breast cancer, increased transcriptional activity of Myc and E2F are characteristic of the basal-like subtype, a tumor subtype with a poor prognosis that originates from an immature progenitor and represents cells arrested at an early stage of differentiation [27,28]. According to our results, BUB1 is among the genes whose expression is increased most in the basal subgroup. This mitotic checkpoint serine/threonine kinase plays a key part in chromosome segregation and cell division, and its up-regulation has been associated with aggressive tumor biology and poor survival. Inhibition of BUB1 in hepatocellular carcinoma cells can prevent tumor proliferation and increase cell apoptosis by regulating the TGF-β/Smad signaling pathway [29]. Moreover, in vitro studies in breast cancer cell lines demonstrated that BUB1, like other mitotic regulators such as AurA and Plk1, is involved in the maintenance of CSC and represents a target for developing anti-CSC therapies [30]. Together, these similarities provide a working hypothesis for future characterization of the stem characteristics of basal ONB, possibly oriented towards the identification of its cell of origin. The recent mapping of the multiple lineage trajectories of olfactory stem cells to the resolution of a single cell [31] could provide an invaluable contribution in this direction. Besides therapeutic implications, the different expression of BUB1 in basal ONB may represent a potential diagnostic indicator of this molecular type that warrants further confirmation. 

We acknowledge that our analysis has some limitations. First, we were not able to identify any clinical or pathological factors linked to prognosis among those described in the literature. This could be due to the relative paucity of patients, but it underlines the importance of the genomic pathways identified, showing their strong prognostic value. Moreover, we limited the analysis to gene expression pathways of the primary tumor, without investigating the mutational status of single genes. 

## 5. Conclusions

In conclusion, we analyzed the transcriptomic profile of ONB, identifying pathways related to worse prognosis and possibly paving the way for new treatment approaches to be integrated into the therapeutic strategy presently used against this rare disease. The generation of reliable gene expression data from FFPE samples, which would allow easy access to hospital pathology archives, with the advantage of being associated with very extended patient survival, is an undoubted strength of this study. A multinational prospective effort is clearly required to put together larger series with the ambitious goal of improving the oncologic outcomes of ONB.

## Figures and Tables

**Figure 1 cancers-13-02527-f001:**
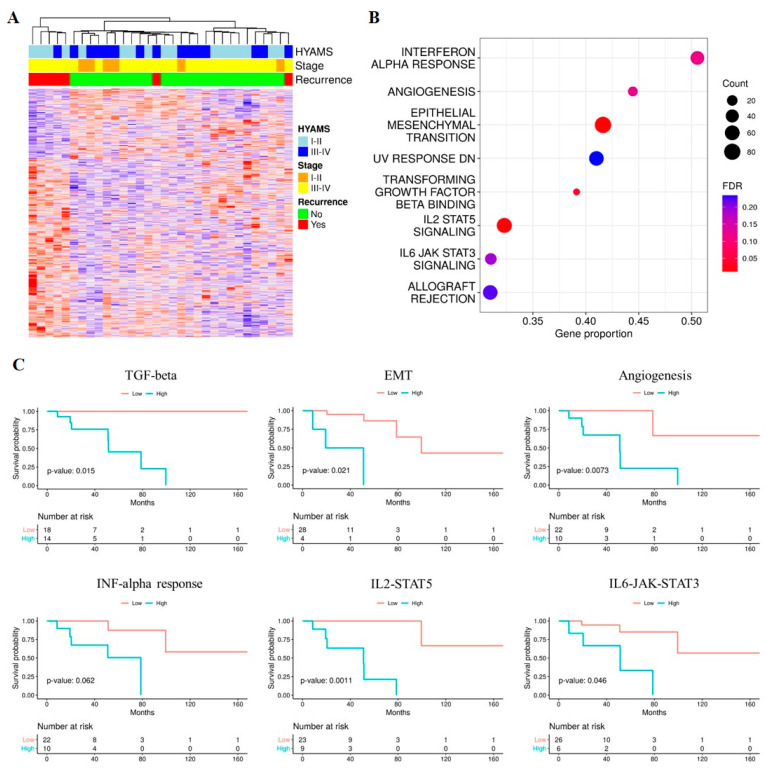
Prognostic analysis for ONB patients. (**A**) Heatmap for hierarchical clustering of relevant genes selected based on DFS hazard-ratio (unadjusted *p*-value < 0.05). Each column represents one ONB sample and each row represents one gene. The red and blue colors indicate higher and lower gene expression, respectively. The upper panel shows clinicopathologic tumor features. (**B**) Functional enrichment analysis depicted biological pathways associated with poor prognosis in ONB tumors. The size of the dots represents the number of genes in the gene set and the color of the dots represents the FDR. (**C**) Kaplan-Meier curves comparing DFS for ONB patients dichotomized according to their pathway-specific global score. *p* values were computed using a log-rank test and adjusted for multiple testing.

**Figure 2 cancers-13-02527-f002:**
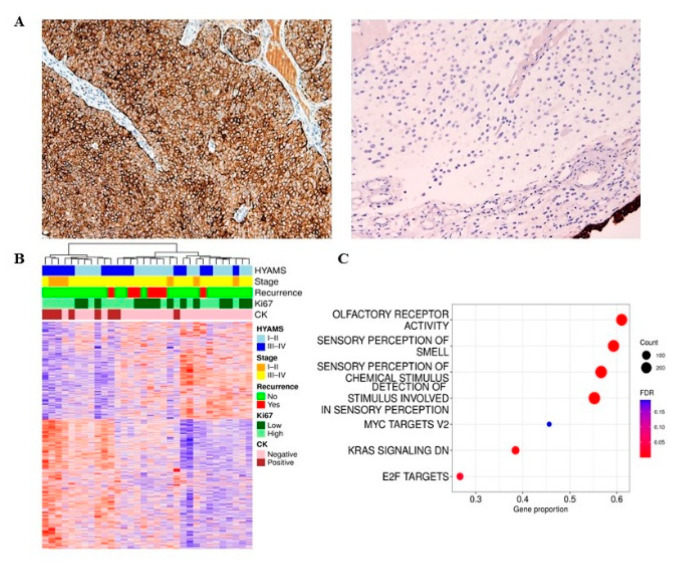
Molecular features of CK^positive^ ONBs. (**A**) Representative immunohistochemical profile of cytokeratin in ONB (10× magnification). Diffuse and strong expression in one CK^positive^ high-grade tumor (left panel) in contrast with a CK^negative^ low-grade tumor (right panel). (**B**) Heatmap and hierarchical clustering analysis of top differentially expressed genes between CK^positive^ and CK^negative^ ONB samples. Supervised clustering clearly separates the CK^positive^ samples from the CK^negative^ ones and shows a distinct heatmap pattern for up- and down-regulated genes. (**C**) Functional enrichment analysis depicted biological processes upregulated in CK^positive^ ONB tumors. The size of the dots represents the number of genes in the gene set and the color of the dots represents the FDR.

**Table 1 cancers-13-02527-t001:** Demographic, clinical and treatment characteristics of ONB patients.

Clinical Variables	N (%)
Gender	
Male	17 (53.1%)
Female	15 (46.9%)
Age at surgery	
median (IQR)	53.5 (20.5)
Surgery	
ER	3 (9.4%)
ERTC	23 (71.8%)
CER	3 (9.4%)
Missing data	3 (9.4%)
Surgical margins	
Negative	19 (59.4%)
Positive	8 (25.0%)
Missing data	4 (12.5%)
pT category–TNM 8th edition	
T1	2 (6.3%)
T2	4 (12.5%)
T3	9 (28.1%)
T4a	0 (0.00%)
T4b	17 (53.1%)
Nodal metastasis	
Absent	31 (96.9%)
Present	1 (3.1%)
Staging—TNM 8th edition	
Stage I	2 (6.3%)
Stage II	4 (12.5%)
Stage III	9 (28.1%)
Stage IV	17 (53.1%)
Adjuvant treatment	
Not performed	5 (15.6%)
Adjuvant RT	22 (68.8%)
Adjuvant ChRT	5 (15.6%)

ER = endoscopic resection; ERTC = endoscopic resection with transnasal craniectomy; CER = combined cranio-endoscopic resection; RT = radiotherapy; ChRT = concurrent chemoradiation.

**Table 2 cancers-13-02527-t002:** Pathological and immunohistochemical features of ONB.

Variables	N (%)
Hyams’ grading classification	
I	1 (3.1%)
II	16 (50.0%)
III	12 (37.5%)
IV	3 (9.4%)
Ki-67 median expression% of positive cells (range)	12.5 (1–75)
Chromogranin expression	
<10%	2 (6.3%)
10–60%	0 (0%)
60–80%	8 (25.0%)
>80%	22 (68.7%)
Synaptophysin expression	
<10%	2 (6.3%)
10–60%	0 (0%)
60–80%	0 (0%)
>80%	30 (93.7%)
Pan-cytokeratin (CK) expression	
Absent	24 (75.0%)
Present	8 (25.0%)
S-100-positive sustentacular cells	
<10%	1 (3.1%)
10–30%	10 (31.2%)
30–80%	2 (6.3%)
>80%	19 (59.4%)
S-100-positive cells inside tumor nests	
Absent	30 (93.7%)
Present	2 (6.3%)

## Data Availability

The microarray data generated in this study are available on request.

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
