# Peer review of "Gene Expression Profiling of Olfactory Neuroblastoma Helps Identify Prognostic Pathways and Define Potentially Therapeutic Targets"

_cancers, 2021, doi:10.3390/cancers13112527_

Round 1

Reviewer 1 Report

Romani et al descibe clinical and molecular analysis of 32 ONB patients/tumors. The Affymetrix profiling performed shows pathway differences in the primary tumors that are not related to current classifications. A number of pathways were identified to predict poor prognosis.

However missing data makes the evaluation of the relevance of the pathways difficult

The group sizes of the high-and –low classes is missing in figure 1c.

An additional factor that could affect these plots is inclusion of patients that have not obtained adjuvant treatment. Several studies showed that OBN is a chemosenstive tumor (Su et al. Head & Neck 39, 2017; Mishima et al. Cancer 101, 2004). It is unclear if patients that do not survive are among the 5 that did not get adjuvant treatment.

Therefore the prediction of pathways might predict future targets, but the proposed improvement in prognostification (title) is premature.

The supervised clustering of CK+/- shows that the 32 ONB can be classified in basal and neuronal as was reported by Classe et al. However the prognostic value of these 2 classes in the current study is missing. Also the recurrence-track is lacking in fig. 2b. This track might show interesting differences between the groups.

The expression analysis adds to a large number of potential targets for a small group of patients. Experimental support for at least 1 of the pathways (i.e. TGF-beta) would increase the impact of the data.

Author Response

The group sizes of the high-and –low classes is missing in figure 1c.

Following the reviewer request, we added this information in Fig. 1c

An additional factor that could affect these plots is inclusion of patients that have not obtained adjuvant treatment. Several studies showed that ONB is a chemosensitive tumor (Su et al. Head & Neck 39, 2017; Mishima et al. Cancer 101, 2004). It is unclear if patients that do not survive are among the 5 that did not get adjuvant treatment.

Thanks for this comment, that correctly underlines the possible confounding factors in the analysis. We confirm that almost all patients (84.4%) underwent adjuvant treatments, while just one patient received induction chemotherapy. The recurrence rate was 20% in patients not receiving adjuvant treatment vs 15% in patients receiving adjuvant RT.

Therefore the prediction of pathways might predict future targets, but the proposed improvement in prognostification (title) is premature.

Thanks for this suggestion. The title has been softened accordingly in:

“Gene expression profiling of olfactory neuroblastoma helps identifying prognostic pathways and defines potentially therapeutic targets”

The supervised clustering of CK+/- shows that the 32 ONB can be classified in basal and neuronal as was reported by Classe et al. However the prognostic value of these 2 classes in the current study is missing. Also the recurrence-track is lacking in fig. 2b. This track might show interesting differences between the groups.

We thank the reviewer for this valuable suggestion and, accordingly, we added the recurrence-track in the heatmap presented in Fig. 2b. As we argued in the “Discussion” section, the two groups show clinical and pathological differences, although in our cohort we cannot demonstrate the prognostic value of this classification, that remains to be established in large cohorts with more informative follow-up. Our contribution was aimed at proposing a possible IHC marker to differentiate between basal and neuronal ONB, with evident advantage for its translation into clinical practice.

The expression analysis adds to a large number of potential targets for a small group of patients. Experimental support for at least 1 of the pathways (i.e. TGF-beta) would increase the impact of the data.

We fully agree with the reviewer thinking. We acknowledge the importance of independent verification of microarray data with alternative techniques, such as quantitative PCR, to verify the accuracy of transcriptome data.  However, we feel confident that our results are reliable, since a high degree of concordance between the Clariom S assay (that we use for the generation of transcriptome data) and qPCR has been demonstrated, thus limiting the need to repeat experiments (Applied Biosystem. Concordance of transcriptome sequencing, microarrays and qPCR using Ion AmpliSeq, Clariom Assays and TaqMan Assays. Technical Note). While our study represents a first explorative work aimed at defining molecular alterations in ONB, selection of leading genes of pathways of interest for qPCR validation of their expression is planned for the future. Of great interest would be the functional validation of the role of specific pathways (ie TGF-beta) in the progression of ONB, as well as to test the efficacy of specific inhibitors alone or in combination with CT, but the lack of in-vitro and in-vivo experimental models of ONB makes this goal unfeasible at the moment.

Reviewer 2 Report

This is a very well-done study on potential biomarkers that may be associated with prognosis in olfactory neuroblastoma. The findings of increased , TGF-beta binding, EMT, IFN-alpha 45
response, angiogenesis, IL2-STAT5, and IL6-JAK-STAT3 expression in patients with poorer prognosis is of clinical and molecular interest and is relatively novel compared with the p-Akt, p-Erk, p-Stat3, ERCC1, TOPO1, TUBB3 , MRP1, pNTRK, Wnt and cKIT/PDGFRA pathways noted in other studies by Topcagic et al and Peng et al. The statistical analysis and data presentation are concise, clear, and nicely done. I do think the authors undersell the utility of clinical and pathological prognostic factors such as Kadish and Dulguerov staging and Hyams grading as these markers, while not perfect by any means, have been shown to have significant prognostic value in several studies over the past 20 years and have been the workhorse factors in predicting survival in olfactory neuroblastoma patients over the years. The identification of molecular profiles and markers associated with prognosis is important in the treatment of olfactory neuroblastoma, nonetheless.

Author Response

We thank the reviewer for the positive evaluation of our work. 

We agree with the reviewer about the relevant prognostic role of both Dulguerov/Kadish system and Hyams classification. In particular, the latter is currently one of the main factors to evaluate in the decision-making process in ONB. It was not our intention to undersell them. Rather, in our introduction we aimed at highlighting the intrinsic limitations of Hyams grade, i.e. the subjective evaluation in its attribution and the issue of tumor heterogeneity in case it was evaluated on a tumor biopsy. Doing so we wanted to motivate one of the main purposes of the paper, i.e. finding new and objective parameters that could integrate or substitute Hyams grade in discriminating between low-grade and high-grade ONB. In fact, increasing evidence is supporting the concept that low and high grade ONB are different diseases deserving different treatment protocols, and therefore an improvement in their discrimination at diagnosis (i.e., before pathologic analysis of the whole surgical specimen) would be a relevant step forward in the management of this disease.

Reviewer 3 Report

The general comment is that the paper is well written and fluent to read. The authors describe a cohort of 32 adult ONB enrolled in two different Institutions between 2001 and 2019; a complete molecular analysis was conducted to identify possible prognostic factors other than those already kwown. All sections of the paper are well presented and the hypothesis on which the analysis was based is clear, statistical analysis is well structured. In the discussion, the authors speculate on the possible innovative therapeutic approaches on the basis of the results obtained. Furthermore,demostrating a valuable intellectual honesty,they analyze the limits of their results due to the small number of patients enrolled over a long period of time.

Author Response

We thank the reviewer for appreciating our research efforts to discover novel prognostic molecular biomarkers of this rare disease.

Round 2

Reviewer 1 Report

The authors have answered the questions. The title was adapted.

The adjuvant treatment does not influence the analysis. However, this information is not revealed in the paper, leaving room for discussion of the data by the readers.

Functional validation in ONB cell lines like TC-268 or JFEN is still recommended.

Author Response

The adjuvant treatment does not influence the analysis. However, this information is not revealed in the paper, leaving room for discussion of the data by the readers.

The reviewer is right, information regarding the potential influence of adjuvant treatment on data analysis was added in the paragraph 3.1  of the “Results” section.

Functional validation in ONB cell lines like TC-268 or JFEN is still recommended.

We are still in agreement with the reviewer thinking, and we thank for suggesting two OBN cell lines for functional studies. Since they are not commercially available, we have kindly requested them on a collaborative basis. However, we would like to point out that an in-vitro study that investigates the role of specific pathways in the progression of ENB, and possibly tests the efficacy of specific inhibitors alone or in combination with CT, is in itself very far from the objective of this study.  ONB is a rare pathology with heterogenous behaviour which likely reflects different molecular pathways driving tumorigenesis.  Molecular studies could be regarded as the starting point to orientate in vitro studies, previa identification of appropriate cell models for resolution of subtype-specific biologic functions.